# Circular polarization-resolved ultraviolet photonic artificial synapse based on chiral perovskite

Qi Liu[1,4], Qi Wei[1,4], Hui Ren[1], Luwei Zhou[1], Yifan Zhou[1], Pengzhi Wang[1], Chenghao Wang[1], Jun Yin[1] & Mingjie Li [1,2,3] ✉

Circularly polarized light (CPL) adds a unique dimension to optical information processing and communication. Integrating CPL sensitivity with light learning and memory in a photonic artificial synapse (PAS) device holds significant value for advanced neuromorphic vision systems. However, the development of such systems has been impeded by the scarcity of suitable CPL active optoelectronic materials. In this work, we employ a helical chiral perovskite hybrid combined with single-wall carbon nanotubes to achieve circularly polarized ultraviolet neuromorphic vision sensing and imaging. The heterostructure demonstrates long-term charge storage as evidenced by multiple-pulsed transient absorption measurements and highly sensitive circular polarization-dependent photodetection, thereby enabling efficient CPL-resolved synaptic and neuromorphic behaviors. Significantly, our PAS sensor arrays adeptly visualize, discriminate, and memorize distinct circularly polarized images with up to 93% recognition accuracy in spiking neural network simulations. These findings underscore the pivotal role of chiral perovskites in advancing PAS technology and circular polarization-enhanced ultraviolet neuromorphic vision systems.

Human vision system cannot perceive the circular polarization state of light, but some animals have more advanced vision systems that are capable of discriminate circularly polarized light (CPL) for perception and communication[1–3]. For example, the mantis shrimp can distinguish the CPL reflected by the uropod and telson of the other, and this invisible communication greatly avoids species conflict. The CPL perception capability of mantis shrimp is ascribed to its compound eye system with rhabdoms acting as quarter waveplates for CPL sensing (Fig. 1a)[4–6]. Furthermore, human vision is also not sensitive to ultraviolet (UV) light. Nonetheless, many animals and insects (including above mentioned mantis shrimp) possess the ability to perceive and discriminate UV light, which is essential for their various purposes such as finding food and detecting predators. Development of CPL sensitive UV light vision

systems for humans is thus a promising avenue for expanding our perceptual capabilities, and potentially unlocking new applications.

In the field of artificial light sensing, photonic artificial synapse (PAS) device which integrates the light detection, learning, and memory in a single device is promising for the development of advanced neuromorphic vision systems[7,8]. However, most of the reported PAS sensors operate in the visible spectrum range and also lack the ability to detect CPL[9,10]. Therefore, CPL-resolved UV artificial vision system is demanding more potential applications in advanced neuromorphic vision sensors, invisible secure communication, holographic technique, augmented reality and virtual reality, bioimaging, healthcare devices and object identification, animal communication and research[11–14].

[1]Department of Applied Physics, The Hong Kong Polytechnic University, Hung Hom, Kowloon, Hong Kong, China. [2]Shenzhen Research Institute, The Hong Kong Polytechnic University, Shenzhen, Guangdong 518057, China. [3]Photonics Research Institute, The Hong Kong Polytechnic University, Hung Hom, Kowloon, Hong Kong, China. [4]These authors contributed equally: Qi Liu, Qi Wei. ✉e-mail: ming-jie.li@polyu.edu.hk

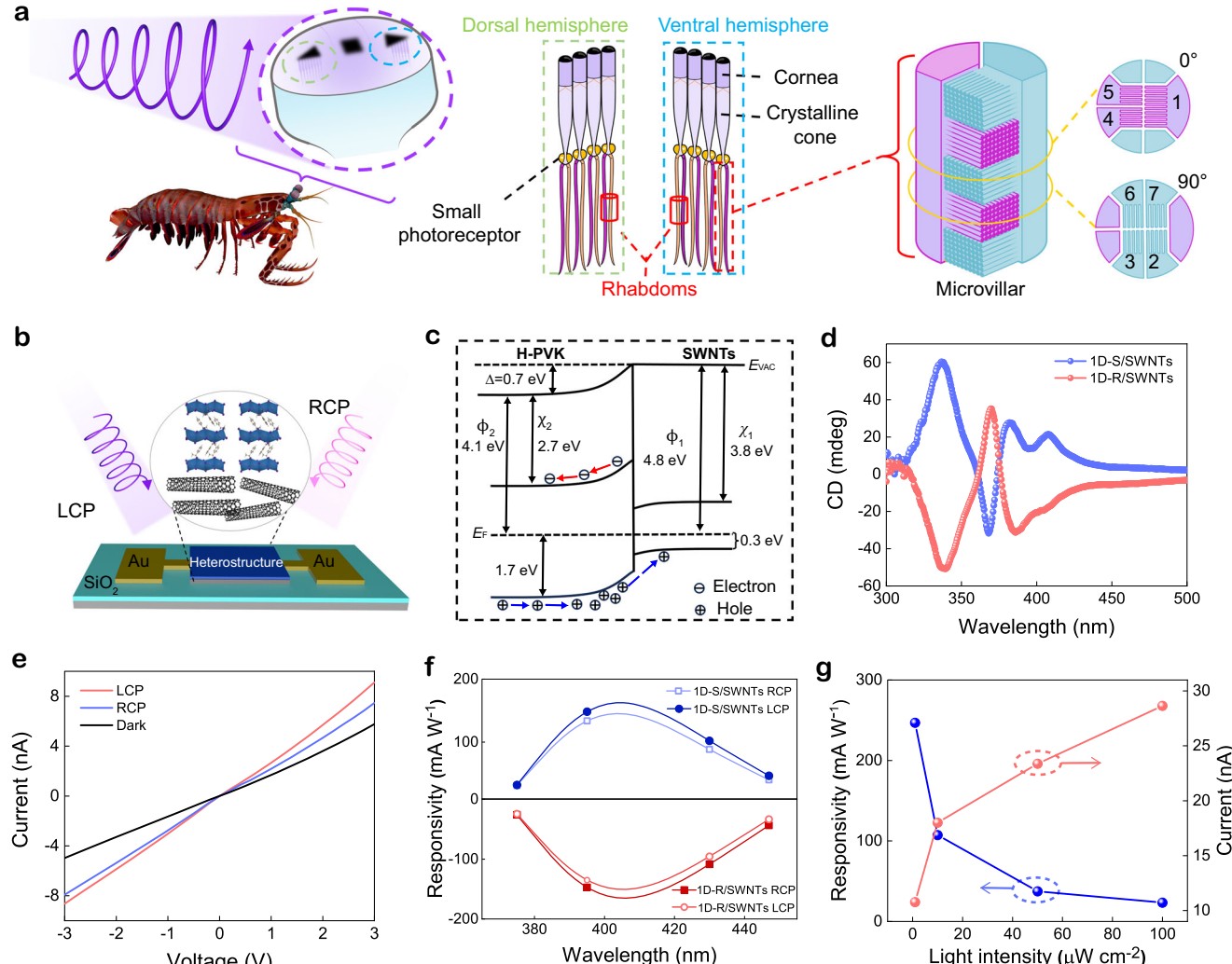

**Fig. 1 | Chiral optoelectronic characterization of perovskite heterostructure.**
**a** Schematic diagram of circular polarization-sensitive vision system of mantis shrimp. The left panel exhibits the frontal view of the compound eye structure composed of the dorsal hemisphere (DH), ventral hemisphere (VH), and midband (MD) between them. The middle panel illustrates the DH and VH structures in detail. The right panel is the microvillar projections from two retinular cells constructing stacked layers in the rhabdom. The enlarged part shows the orientation of microvilli in alternate layers of rhabdom. The mantis shrimp image was rendered from the royalty-free 3D model from Turbosquid. DH, VH, and the enlarged parts are redrawn from ref. 6. **b** Structural scheme of the in-plane 1D-S/SWNTs heterostructure device. **c** Energy band diagram of H-PVK/SWNTs heterostructure ($E_{vac}$ is the vacuum energy level, $E_F$ is the Fermi energy level). **d** CD spectra of H-PVK (1D-R, 1D-S)/SWNTs heterostructures. **e** Current-voltage curves of 1D-S/SWNTs heterostructure under dark, LCP, and RCP (395 nm, 10 μW cm⁻²) illumination. **f** Responsivity of H-PVK/SWNTs under RCP and LCP at the wavelengths of 375 nm, 395 nm, 430 nm and 447 nm (10 μW cm⁻²). **g** Photocurrent and responsivity under the incident 395 nm LCP at different intensities.

Several chiral materials have been developed and used in CPL photodetectors such as chiral organic small molecules, conjugated polymers, and plasmonic metamaterials, however, CPL-resolved PAS has rarely been developed[15–18]. To achieve a CPL-resolved UV PAS device, the functional layers should have not only handedness-sensitive UV optical absorption, and efficient charge transport, but also synaptic behaviors simultaneously. In recent years, solution-processed halide perovskites have demonstrated remarkable performance in low-cost efficient optoelectronics and there has been growing interest in the development of chiral perovskites for applications in chirality-related fields, such as chiral emission, ferroelectrics, memory and spintronics[19–21]. Chiral perovskites with left and right-handedness can be effectively developed from the intentional introduction of chiral molecules such as α-Methylbenzylamine enantiomers (α-MBA), β-Methylphenethylammonium (β-MPA), 1-Cyclohexylethylammonium (CHEA), etc[22–25]. These chiral perovskites thus have potential in sensing CPL and differentiating between left-handed CPL (LCP) and

right-handed CPL (RCP)[26–29]. Specifically, helical one-dimensional chiral perovskite (H-PVK) exhibits relatively high circular dichroism (CD) owing to the transferred chirality by the strong electronic interaction between chiral ligands and $BX_6^{4-}$ matrix (B = Pb, Sn, Ge; X = I, Br, Cl)[30–32]. However, the current H-PVK materials suffer from low conductivity, which discounts their electrical performance in optoelectronics. Combining H-PVK with high-mobility semiconductors could promote the transfer of photoinduced carriers and circumvent poor carrier transport[33,34]. Therefore, exploration of H-PVK-based heterostructure and their underlying CPL-dependent carrier dynamics and photoresponse performance is essential in developing CPL-resolved PAS devices.

In this work, we develop the H-PVKs/single-wall carbon nanotubes (SWNTs) heterostructure to achieve the CPL-modulated UV PAS devices. The heterostructures exhibit high circular polarization-dependent photoresponsivity to discriminate LCP and RCP beams. Ultrafast, efficient, and circular-polarization dependent hole transfer from

H-PVKs to SWNTs contributes to the circular-polarization resolved photocurrent with a photoresponsivity up to 240 mA W⁻¹. A spike-timing-dependent transient absorption (TA) measurement is designed to monitor the carrier populations to reveal the chiroptical synaptic properties of the heterostructure. A series of polarization-dependent synaptic activities are further demonstrated on the PAS device with successful CPL perception, learning, and recognition behaviors. More importantly, the PAS-based spiking neural networks (SNNs) are simulated and exhibit a high accuracy up to 93%. The proposed heterostructure PAS exhibits great potentiality in building CPL-modulated neuromorphic vision systems with high biological plausibility, giving rise to new paradigms in various optoelectronic applications.

## Results

### Circular polarization-resolved UV photodetection

To fabricate the CPL sensitive absorber in the PAS device, chiral perovskite H-PVKs, and their heterostructures with SWNTs layer are prepared (see Methods). Here, one-dimensional (1D) chiral halide perovskite is selected due to its high chiroptical performance in the UV region[35,36]. SWNTs, which has high mobility and is widely used as hole extraction layer in perovskite solar cells[37,38], is chosen as the charge-transporting layer in our devices. Chiral 1D (*S*-α-MBA)PbI₃ (abbreviated as 1D-S) and (*R*-α-MBA)PbI₃ (abbreviated as 1D-R) are synthesized using left-hand (*S*-α-MBA) and right-hand (*R*-α-MBA) enantiomers, respectively (see schematic crystal structures in Supplementary Fig. 1). The crystal structure, morphology, and thickness of H-PVK thin films (~70 nm) as measured by X-ray diffraction, surface profiler and atomic force microscope (AFM) measurements, respectively indicate the high quality of 1D H-PVK film are achieved (Supplementary Figs. 2–4). Both the 1D-R and 1D-S perovskites show intense excitonic absorption at 375 nm (Supplementary Fig. 5) but an opposite CD signal at the same wavelength, indicating the difference between the two kinds of heterojunction comes from corresponding *S* and *R* enantiomers of MBA. The 1D-R and 1D-S films demonstrate high discrimination with optical anisotropy factors of CD ($g_{CD}$) of 0.01 at around 340 nm transitions (Supplementary Fig. 6).

A conjugated polymer wrapping sorting method is developed to obtain the semiconducting SWNTs (Supplementary Fig. 7)[39]. Raman spectroscopy, AFM (Supplementary Fig. 8 and 9) results reveal the large-area, smooth SWNTs film can be deposited on Si wafer serving as charge transporting and storage media in our designed PAS device. Figure 1b shows the sketch of the two-terminal PAS device with the corresponding fabrication process shown in Supplementary Fig. 10. Briefly, the SWNTs film is deposited by soaking the substrate (SiO₂/Si) in the semiconducting SWNTs solution followed by a heating process. Next, the electrodes (45 nm Au/5 nm Ti) are developed by photolithography, electron-beam evaporation, and a lift-off process. Optical and scanning electron microscope images demonstrate the good uniformity of patterned electrodes with sparsely distributed SWNTs in the channel (Supplementary Fig. 11). After the deposition of the electrode, the H-PVK film is spin-coated onto the patterned substrate followed by an annealing process. The band alignments of materials are determined by ultraviolet photoelectron spectroscopy together with UV-Vis measurements (Supplementary Fig. 12)[40]. Upon contact between H-PVK and SWNTs, an interfacial band bending with a potential barrier of 0.7 eV is formed (Fig. 1c), which can repel the photoexcited electrons and facilitate the transfer of photoexcited holes from H-PVK to SWNTs (see details in Supplementary Fig. 13). More evidence of photoexcited charge separation are provided in the following section. Notably, 1D-R/SWNTs and 1D-S/SWNTs heterostructures exhibit obvious symmetrical CD signals in UV region (Fig. 1d), and no CD signal is observed in pristine SWNTs network (Supplementary Fig. 14), which indicates that the chirality mainly originates from 1D-S and 1D-R. Such chiral 1D-R/SWNTs and 1D-S/SWNTs

heterostructures are thus utilized as CPL-sensitive materials in the following chiroptoelectronic and neuromorphic vision device development.

To examine the chiroelectronic performance, the photoresponse of the heterostructure is measured under CPL illumination with different handedness, wavelengths, and intensities. As shown in Fig. 1e, the 1D-S/SWNTs device exhibits obviously distinguishable photoresponse under two kinds of CPL illumination, and the photocurrent is larger than the other reported chiral perovskite-based photodetectors[41], contributing to high-performance chiroptical electronics. The photoresponsivity under CPL illumination at different wavelengths is determined and depicted in Fig. 1f with voltage-current curves shown in Supplementary Fig. 15. A relatively larger responsivity with a peak at around 400 nm is observed, which is consistent with the chiroptical absorption features of the H-PVK layer. The light intensity-photoresponse results are shown in Fig. 1g, and the 1D-S/SWNTs device exhibits a high responsivity above 240 mA W⁻¹ under LCP illumination with an incident power of 1 μW cm⁻², suggesting a superior capability for sensing weak CPL. In comparison, the 1D-R/SWNTs device demonstrates similar photoelectric performance but higher responsivity under RCP illumination. Besides, the constructed heterostructures show high stability and negligible degradation after ambient storage for over one month (Supplementary Fig. 16). These chiroptoelectronic characteristics of our chiral perovskite heterostructure device thus indicate the great potential in the application of CPL-resolved vision.

### Mechanism of CPL-dependent carrier dynamics and storage

To gain insight into the photophysical mechanism of our CPL-sensitive H-PVK/SWNTs heterostructure device and understand its protentional for use in neuromorphic vision with photosynaptic activity, ultrafast polarization-dependent transient absorption (TA) in UV and near-infrared (NIR) are performed. Figure 2a shows the schematic diagram in which CPL serves as a pump signal and linear polarized light is utilized to probe the carrier population dynamics. Under optical excitation, the photoexcited holes can be transferred from H-PVK to the surface-attached SWNTs. Figure 2b and Supplementary Fig. 17 sketch the established molecule model of 1D-S/SWNTs for and density functional theory (DFT) calculations. Figure 2c shows the calculated projected density of states (PDOS) of SWNTs and 1D-S perovskite. The HOMO of SWNTs is higher than the VBM of the 1D-S, which is consistent with Fig. 1c and thus promotes the photoexcited hole transfer toward the SWNTs layer. Meanwhile, a similar charge transfer process is demonstrated in 1D-R/SWNTs heterostructure (Supplementary Fig. 18). Figure 2d shows the pseudo-colour plot of TA spectra of 1D-S and 1D-S/SWNTs excited by 340 nm LCP femtosecond (fs) laser beam. Upon photoexcitation, a photobleaching (PB, i.e., negative TA signal $\Delta A$) band at around 375 nm is observed due to the state filling of excitonic states in H-PVK. 1D-S perovskite exhibits a long exciton lifetime ($\tau_{1D-S}$) of more than 16 ns. For 1D-S/SWNTs heterostructure, an obvious PB quenching along with an additional fast decay with lifetime ($\tau_{1D-S/SWNTs}$) of around 2 ns are observed (Fig. 2e and normalized TA dynamics in Supplementary Fig. 19). Note that PB signal is proportional to the sum of the populations of photoexcited electrons ($\Delta n_e$) and holes ($\Delta n_h$) involved in the transition (i.e., $\Delta A \propto (\Delta n_e + \Delta n_h)\alpha_0$, where $\alpha_0$ is linear absorption coefficient). The reduction of PB signal is thus ascribed to the photoexcited hole transfer from 1D-S into the SWNTs network. The hole injection efficiency ($\eta_i$) from 1D-S to SWNTs is estimated to be ~72% (by using $\eta_i = 1 - \tau_{1D-S/SWNTs}/\tau_{1D-S}$) under LCP illumination[42]. Compared with LCP excitation, RCP result in a smaller $\eta_i$ of 60 % due to the lower photoresponse of RCP by 1D-S perovskite. The pseudo-colour plots of TA spectra of 1D-S and 1D-S/SWNTs excited by RCP beam is shown in Supplementary Fig. 20. Additionally, 1D-R/SWNTs heterostructure demonstrates a higher $\eta_i$ under the RCP excitation (Supplementary Fig. 21).

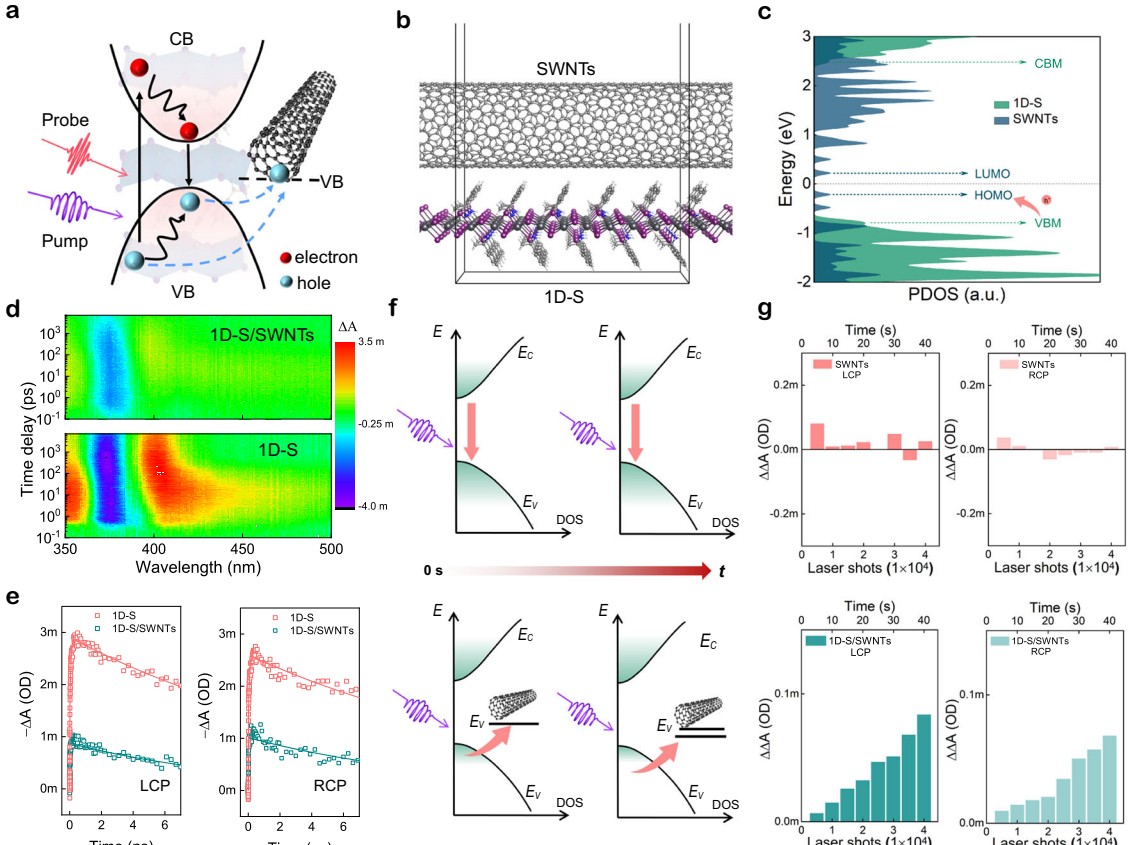

**Fig. 2 | CPL-dependent charge transfer dynamics and storage. a** Schematic illustration of carrier dynamics in H-PVK/SWNTs heterojunction. CB, conduction band; VB, valence band. **b**, **c** Optimized interfacial structures, and corresponding PDOS of 1D-S/SWNTs. CBM, conduction band minimum; VBM, valence band maximum. The arrow indicates the photoexcited hole transfer from VBM of H-PVK to HOMO level of SWNTs. **d** Pseudo-colour plot of TA spectra of H-PVK and 1D-S/ SWNTs measured under 340 nm LCP excitation. **e** Band edge ground-state bleaching (GSB) dynamics of 1D-S and 1D-S/SWNTs under 340 nm LCP/RCP excitation. **f** Band structure variations versus CPL illumination time of H-PVK (upper panel) and H-PVK/SWNTs (lower panel). **g** Laser spike-numbers dependent change of TA signal amplitudes probed at -1050 nm and 1 ps of SWNTs (upper panel) and 1D-S/SWNTs (lower panel) under 340 nm LCP/RCP excitation.

Figure 2f illustrates the energy band variation of the H-PVK and heterostructure under multiple CPL pluses illumination. For the H-PVK alone, when the CPL beam is off, the excited excitons recombine rapidly within its lifetime of tens of ns (upper panel left). Therefore, there is no difference for the samples when the next CPL spike arrives (upper panel right). For the heterostructure, the excitons can be effectively dissociated at the interface after excitation, where electrons are captured by the trap states inside the H-PVK and holes are transferred into SWNTs with reduced possibility for recombination. The NIR TA is used to probe the carrier population dynamics in SWNTs. A long-lived PB band at -1050 nm is observed in H-PVK/SWNTs, which is attributed to the photoexcited hole-transfer from H-PVK to SWNTs (see details in Supplementary Fig. 22). Consequently, the extracted holes density could increase in SWNTs after multiple-pulse excitation in the heterostructure (lower panel of Fig. 2f), resulting in the increased photocurrent for the PAS application.

To prove the above assumption, we monitor the change of TA signal ($\Delta\Delta A$), which reflects the carrier population variations as a function of laser spike numbers. As shown in Fig. 2g, during the stimuli by 1 kHz fs laser pulse spikes, since the fast recombination of excitons inside the SWNTs (upper panel of Fig. 2g), the $\Delta\Delta A$ signal is almost unchanged over time. Meanwhile, the $\Delta\Delta A$ signal in the UV region probed at the band edge of 1D-S (upper panel of Supplementary Fig. 23) also shows no obvious variation, which is identical to the optoelectronic performance of 1D-S based device (Supplementary Fig. 24). In contrast, due to the effective charge separation and storage in the heterostructures, $\Delta\Delta A$ of 1D-S/SWNTs in NIR (lower panel of

Fig. 2g) and UV regions (lower panel of Supplementary Fig. 23) gradually increase with LCP pump spike numbers and have higher values as compared with RCP excitation. This intriguing characteristic therefore can contribute to the development of CPL-resolved artificial synapses with high biological plausibility. The above findings demonstrate the CPL polarization preference photocurrent features as well as the memory functions in our heterostructures for PAS devices development.

## Circular polarization-dependent synaptic activities

Next, a series of retina synaptic plasticity in response to CPL stimuli are imitated by the PAS device. Figure 3a shows the schematic optical setup for circular polarization-dependent synaptic testing (with details of CPL spike generation provided in Supplementary Fig. 25). The photocurrent of 1D-S/SWNTs (Figs. 3b) and 1D-R/SWNTs (Fig. 3c) heterostructure PAS devices have obvious relationship with ellipticity of the incident beam (395 nm, 10 $\mu$W cm$^{-2}$), which thus enables the accurate distinction of incident CPL and synaptic plasticity tunability of PAS from the degree of circular polarization as shown below. Owing to the symmetrical chiroptical features of 1D-S/SWNTs and 1D-R/SWNTs, the CPL-induced synaptic activities of PAS devices based on 1D-S/ SWNTs are presented as a representative in detail in the main text. As shown in Fig. 3d, excitatory postsynaptic current (EPSC) is triggered by a single LCP/RCP spike, and the current gap between peak EPSC values implies that our PAS device can respond differently to CPL stimuli with opposite circular polarizations. 1D-S/SWNTs PAS device demonstrates 1.2 times higher EPSC response under the modulation of LCP spikes as

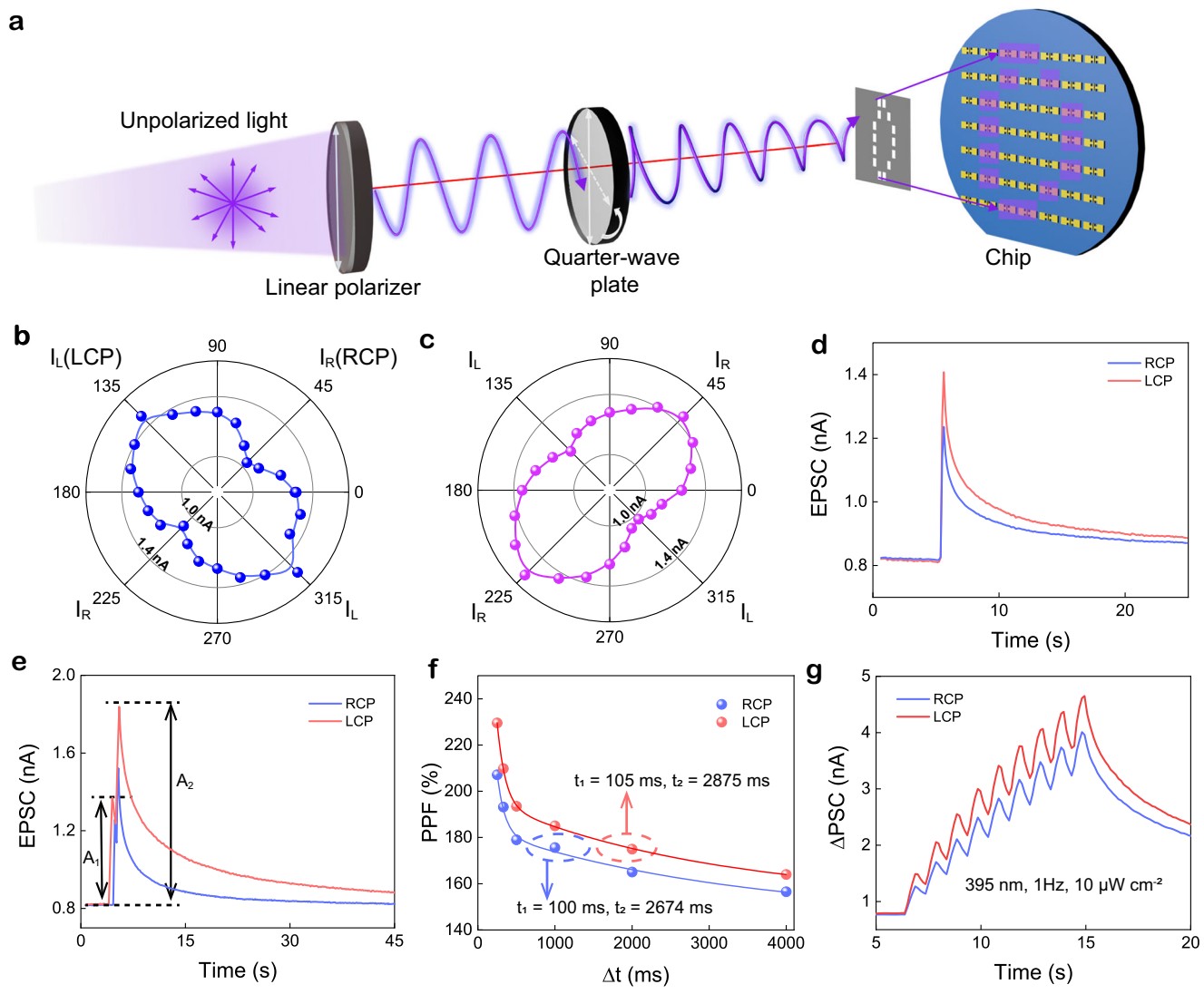

**Fig. 3 | Circular polarization-dependent synapse activities. a** Schematic diagram of optical setup for the CPL sensing and imaging measurement. **b, c** Polar plots of the polarization-dependent response current of 1D-S/SWNTs and 1D-R/SWNTs based PAS, respectively. **d** Transient EPSC of PAS device in response to single LCP and RCP (395 nm, 10 $\mu$W cm$^{-2}$) pulses. **e** EPSC induced by a pair of LCP/RCP spikes (395 nm, 10 $\mu$W cm$^{-2}$) with the interval of 1 s. **f** Variation of PPF index with the increase of excitatory stimuli interval. **g** EPSC of 1D-S based PAS excited by consecutive LCP and RCP spikes.

compared with RCP excitation, demonstrating the CPL-resolved synaptic behavior. Moreover, paired-pulse facilitation (PPF), a critical characteristic of short-term plasticity (STP), is also imitated by PAS device. PPF is related to the temporal encoding/decoding of visual information, which is reflected by an increase in postsynaptic response $A_2$ after the second spike as compared with the response $A_1$ after the initial spike[43,44]. Figure 3e shows the PPF behavior triggered by two CPL spikes with 500 ms duration and 10 $\mu$W cm$^{-2}$ intensity. Note that the increment of $A_2$ typically depends on the time interval ($\Delta t$) between two spikes, which can be described by the PPF index of $A_2/A_1$. A larger PPF index can lead to a higher temporal resolution for efficient and accurate processing of temporal information. As shown in Fig. 3f, PPF index decays slowly with increasing $\Delta t$, and the LCP-induced PPF has a higher PPF index up to around 230% with a slightly longer lifetime (as fitted by PPF $= C_1 e^{-\Delta t/t_1} + C_2 e^{-\Delta t/t_2}$). In addition to STP, the long-term plasticity (LTP) characteristic is investigated by performing the consecutive CPL spikes on PAS device by a train of nine CPL spikes (Fig. 3g). Obviously, 1D-S/SWNTs PAS shows step-by-step increased EPSC value under the consecutive CPL spikes, which is consistent well with above CPL pulse-laser induced carrier increment (Fig. 2g) and indicates an enhanced strength of synaptic plasticity by learning times.

Moreover, CPL at other wavelengths (430 nm and 447 nm) can also induce LTP behaviors (Supplementary Fig. 26) but with smaller differences under different polarization spikes, which is consistent with their lower CD signals.

The synapse plasticity strength of our PAS device can be flexibly tuned by encoding CPL spikes with different numbers, wavelength, intensity, frequency, and polarization, thereby ensuring multiple learning and memory functions. Figure 4a shows that after applying two LCP spikes (1 Hz, 10 $\mu$W cm$^{-2}$) to 1D-S/SWNTs-based PAS, the EPSC strength decays to 1 nA within 60 seconds, indicating short-term potentiation (STP). When the number of stimuli spikes is increased to nine, the decay of EPSC response takes a longer time (120 seconds), which is indicative of long-term potentiation (LTP) behavior. Figure 4b summarizes the CPL spike number-dependent EPSC ratio ($A_n/A_1$). The rate of growth of $A_n/A_1$ slows down and eventually reaches a maximum value of 648%, indicating the transition from STP to LTP, which may be due to the limited hole-trapping sites in SWNTs and carrier lifetimes. Figure 4c shows that the PAS device exhibits peak current responses of 1.75, 2.64, and 4.18 nA under three different wavelengths while the 532 nm cannot excite the PAS device, indicating superior UV plasticity behavior. Figure 4d shows that CPL intensity dependent EPSC behavior

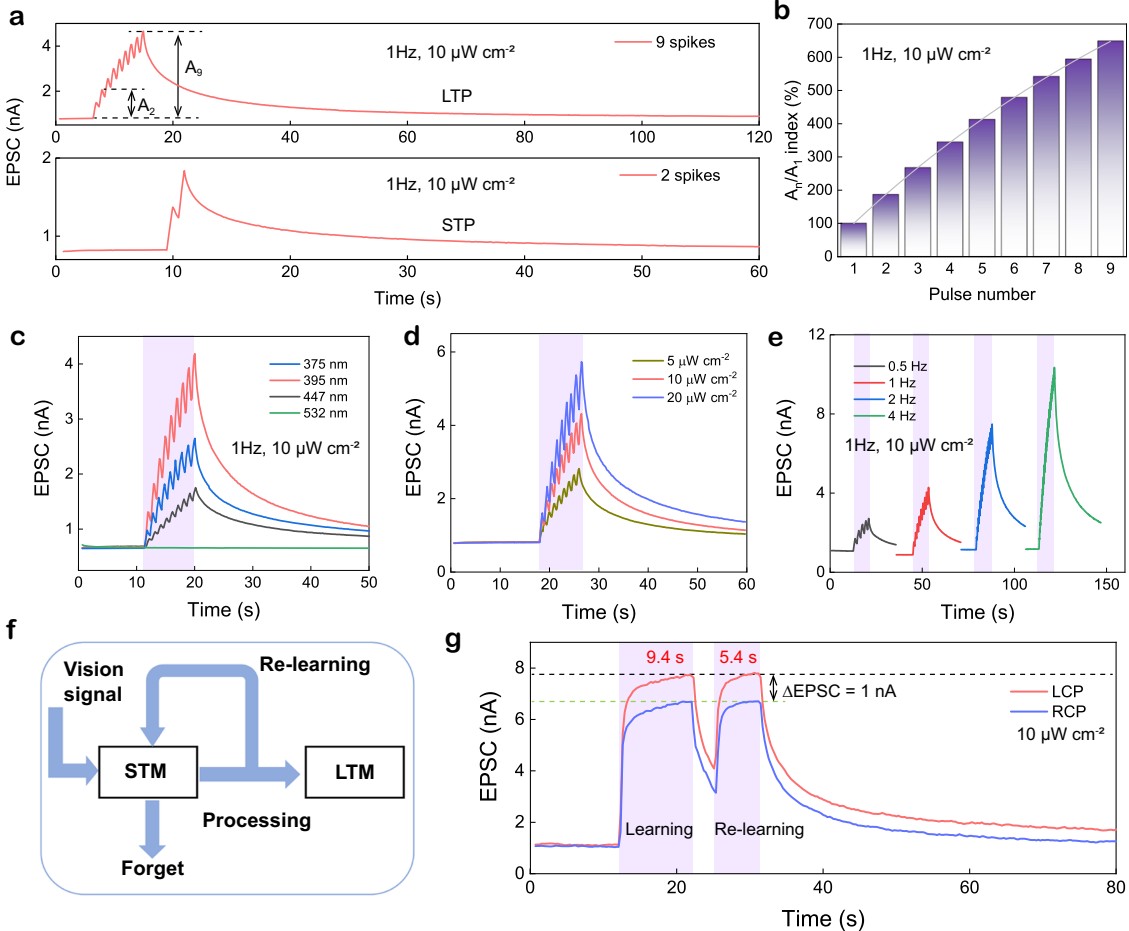

**Fig. 4 | Tunable CPL synaptic characteristics. a** LTP/STP behaviors under 9 and 2 LCP spikes. **b** Variation of $A_n/A_1$ index with increasing the LCP pulse numbers. **c** EPSC in response to multiple LCP pulses with various wavelengths. **d** EPSC in response to 395 nm LCP with different power intensities and fixed frequency of 1 Hz. **e** Spike-frequency-dependent EPSC triggered by a sequence of 9 LCP spikes. **f** Learning and memory process in neuromorphic vision system. **g** Learning-experience behavior imitated by the PAS device under 395 nm LCP and RCP illumination. The sample is 1D-S/SWNTs-based PAS.

of stronger EPSC response with slower decay at higher intensity. Furthermore, in spike-frequency-dependent plasticity (SFDP), a larger EPSC value with slower decay can be observed at high spike frequencies (Fig. 4e), which is consistent with the fact that multiple stimuli in a short time can enhance learning and memory in organisms.

The relationship between memory retention and time corresponds to the Ebbinghaus forgetting curves[45,46], and repeating the learning process accompanied by forgetting experiences (Fig. 4f) can convert short-term memory (STM) to long-term memory (LTM). In this experiment (Fig. 4g), the gray dash line represents the learning target level, and the PAS is exposed to two types of 395 nm CPL for studying polarization-dependent learnability. Under LCP illumination, it takes 9.4 seconds for the 1D-S/SWNTs PAS device to reach the target level initially, while subsequent re-learning processes only require 5.4 seconds to recover all the cognition. However, under the RCP with the same intensity and $\Delta t$, the PAS device shows a relatively low EPSC value (green dashed line) and weak learnability. Further, the difference in learning and memory between LCP and RCP stimuli is investigated in Supplementary Fig. 27. It can be observed that the 1D-S/SWNTs structured PAS demonstrates better learnability under LCP illumination.

### CPL image recognition and memory

The PAS device with CPL-resolved perception, learning, and memory capabilities enables image recognition and spiking neural networks (SNNs) for deep learning in artificial neuromorphic vision systems.

Figure 5a describes the CPL image vision process inspired by the optical communication between mantis shrimps via reflection of CPL from their cuticle[47]. Vision signals (the telson keel outline in 7 × 7 pixels with measurement configuration schematically show in Fig. 3a) are encoded as CPL spikes with varying spatiotemporal information to program the developed array. The color intensity of each pixel is represented by the measured EPSC value in each PAS device in the array. The image recognition involves the variation of color levels for each pixel over LCP, RCP, and linearly polarized light pulse numbers in the mapping (Fig. 5b and Supplementary Fig. 28). The synapse weight (EPSC values) of the PAS array exhibits enhanced sharpness after applying a sequence of stimuli (3, 5, and 7 spikes), and memorization becomes gradually clear. It is worth noting that the LCP-induced memorized graphic exhibits higher EPSC values and a recognizable likeness to the input pattern, indicating that this mode is the most effective visual perception method among the three image recognition processes. All perception results suggest that our PAS device has great potential for constructing polarization-dependent neuromorphic vision systems.

Lastly, to further assess the potential of the PAS in large-scale image recognition tasks, SNNs are simulated with PAS modeling as spiking neurons[48]. As shown in Fig. 5c, the SNNs architecture consists of 784 input, 200 hidden, and 10 output neurons. In the simulation, the input dataset is converted to spikes and transmitted to the first layer, where each neuron corresponds to one pixel of the image dataset. During the feedforward process, the spiking neuron accumulates the

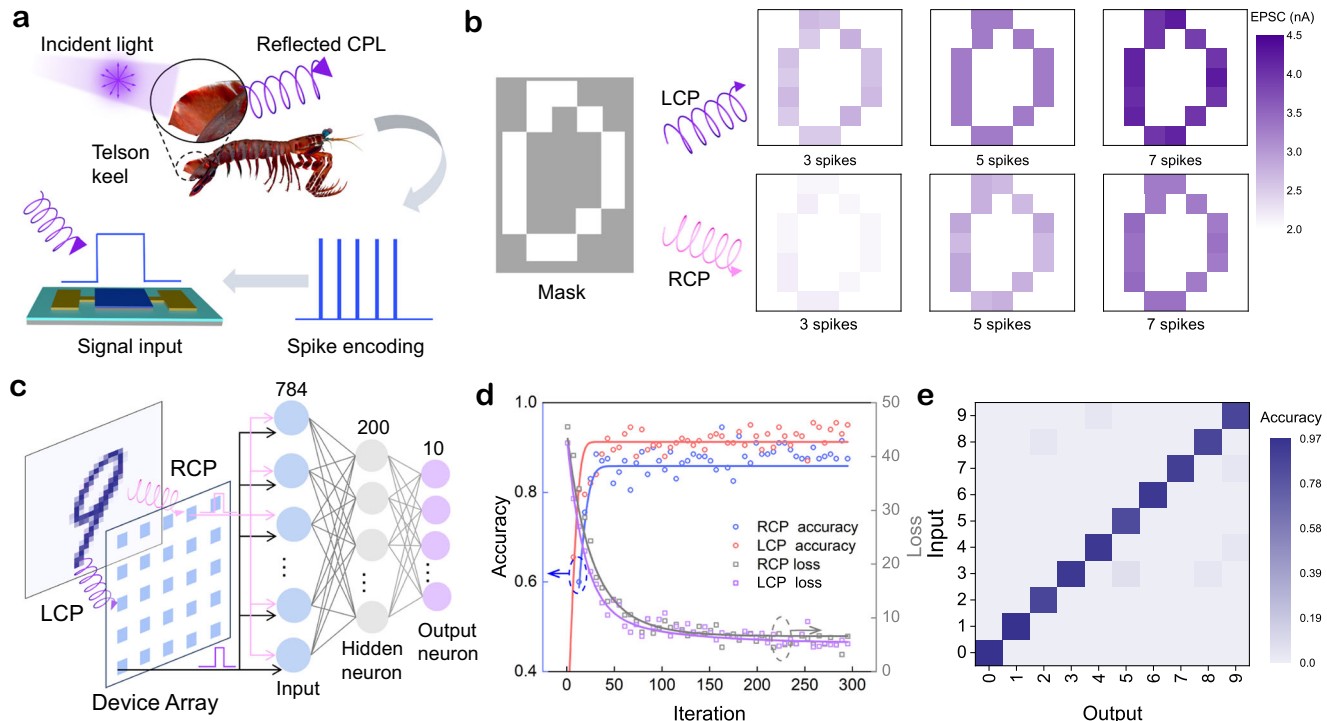

**Fig. 5 | CPL image memory and recognition. a** Schematic diagram of CPL imaging. **b** Object and measured weights of the object images in the initial state after a sequence of 395 nm LCP and RCP spikes (10 $\mu$W cm$^{-2}$, 1 Hz) by the PAS array with schematic experimental setup shown in Fig. 3a. **c** Schematic structure of PAS device based neural network. **d** CPL-dependent recognition accuracy and loss variation of the SNNs during 300 training epochs. **e** Recognition accuracy to every target after the training of SNNs under LCP spikes.

weighted sum of input spikes and generates the membrane potential of the neuron. When the input provides sufficient excitation, the membrane potential reaches a threshold ($\theta$), and the neuron releases a spike to its next connections[49]. Additionally, the backpropagation algorithm is employed for weight updates, with the released activation value of the current layer serving as the input for the subsequent layer (see computation process details of the SNNs in Supplementary Fig. 29). The Modified National Institute of Standards and Technology (MNIST) database is selected as the training and test target for the evaluation. Specifically, the recognition accuracy of SNN model is evaluated using 60,000 training and 10,000 test images with a batch size of 200 at each epoch. Figure 5d shows the test results with an increasing recognition accuracy and decreasing loss value over iteration times. After 300 epochs, the LCP and RCP-modulated SNNs achieve recognition accuracy of ~0.93 and ~0.88 respectively, exhibiting polarization-dependent learning efficiency. Furthermore, the recognition results for LCP-based test are summarized in Fig. 5e. Each test set achieves high accuracy, indicating that the PAS-based SNNs have superb cognitive and classification capabilities for test data after training. To the best of our knowledge, these PAS devices demonstrate the first efficient UV-CPL-dependent synaptic behaviors and optical imaging for artificial neuromorphic vision system development (Supplementary Table 1).

## Discussion

In summary, we have developed a UV-CPL sensitive heterostructure and utilized it as a chiroptoelectronic converter for polarization-dependent neuromorphic electronics. The heterostructure is composed of H-PVK with circular dichroic response ability and SNWTs with good conductivity, achieving a high photoresponsivity above 240 mA W$^{-1}$ at 395 nm. As a CPL-resolved PAS device, this heterostructure successfully imitates a series of synaptic activities (e.g., EPSC, PPF, SFDP, STP, and LTP) by modulating input LCP/RCP beams

encoded with different spatiotemporal information. Moreover, the PAS device is assembled as an array, and achieves circular polarization-dependent image perception, learning, and recognition, which shows great potential in polarized imaging and optical communication. Notably, such PAS is utilized to establish the SNNs, and reaches a high recognition accuracy of up to 93% in pattern recognition. Such chiral perovskite heterostructure as circular polarization-resolved prototypes shows great prospects in constructing intelligent neuromorphic vision systems.

## Methods

### Materials

R-(+)-α-Methylbenzylamine (≥99.0%), S-(-)-α-Methylbenzylamine (≥ 99.0%), N,N-dimethylformamide (DMF, 99.5%), Toluene (anhydrous, 99.8%), Fifty-seven percent aqueous and hydriodic acid (HI) solution (99.95%) were purchased from Sigma-Aldrich. Lead oxide (PbO, 99.9%) was purchased from Alfa Aesar. The raw arc-discharge CNT (AP-SWNT, ≥ 60%) was purchased from Carbon Solution Inc. The dispersant 9-(1-octylonoyl)-9H-carbazole-2,7-diyl (PCz) was prepared by Suzuki poly-condensation in a relatively high yield.

Synthesis of (R- and S-α-MBA)I precursors: The precursors (R-α-MBA) I and (S-α-MBA)I were synthesized by mixing HI with the chiral amines (R- and S-MBA) in 1:1 molar ratio. First, 5.3 ml (0.0416 mol) of ammine was dissolved in 20 ml methanol, to which the 5 ml (0.0416 mol) HI was added dropwise within 15 min, and the mixture was stirred at 0 °C for 2 hours. Then the solvent was removed by rotary evaporation to yield the (R- and S-α-MBA)I white powder. Finally, the powder was collected and washed with diethyl ether followed by drying in the vacuum overnight.

Synthesis of H-VPK: The (R-/S-α-MBA)I templated by different α-methylbenzylamine enantiomers and PbI$_2$ with an equal molar ratio were dissolved into DMF to obtain the 1D-R/1D-S, and the solution concentration is controlled at 0.3 mol L$^{-1}$.

## Sample characterization

X-ray diffraction characterization was performed by Rigaku SmartLab with monochromatized Cu Kα radiation ($\lambda = 1.54$ Å). The surface morphology of the PAS device was characterized by a field-emission scanning electron microscope (Tescan MAIA3) and Scanning Probe Microscope (Asylum MFP-3D Infinity). The thickness of the H-PVK was measured by Bruker DektakXT Surface Profiler. The ultraviolet photoelectron spectroscopy data was measured by a Thermo Scientific Nexsa instrument.

## Optical spectroscopy

TA spectroscopy was measured using an amplified titanium: sapphire femtosecond laser (central wavelength 800 nm, pulse width 50 fs, repetition rate 1 kHz, Coherent Libra) coupled with an optical parametric amplifier (pump wavelength tunable from UV to IR, OPerA Solo) and a Helios pump-probe setup (probe wavelength tunable from UV to NIR, Ultrafast Systems). Raman spectra were obtained using the WITEC alpha300 R- Raman imaging microscope. CD spectra was characterized by a CD spectrometer (JASCO J-1500 Easton, MD, USA) at room temperature with a data pitch of 1 nm and a scan rate of 100 nm per minute.

## Device fabrication

The wafer with a 300 nm thick oxide layer was utilized as substrate and was washed ultrasonically with acetone, ethanol, and distilled water for 15 min in turn. After that, the substrate was treated with UV-ozone plasma for 10 min. The SWNTs film was deposited by soaking the substrate into the semiconducting SWNTs solution for 24 h, followed by heating at 120 °C for 15 min. The electrodes (5 nm Ti/45 nm Au) were defined by typical photolithography, electron-beam evaporation, and a lift-off process. Thin chiral perovskite film was obtained by spin coating the H-PVK solution onto the patterned substrate with a spin rate of 4000 rpm for 30 s, followed by annealing at 95 °C for 15 min.

## Device characterization

The electrical measurement of PAS devices was performed in the Lake Shore probe station equipped with a semiconductor analyzer (Keithley 4200-SCS), and the reading bias is 0.5 V. CPL was generated by using a linear polarizer (WP25M-UB), quarter waveplate (AQWP10M-340) and LEDs with different wavelengths purchased from Thorlabs. The intensity of CPL was calibrated by a standard optical power meter (Newport 843-R with a PD300-UV optical power detector) before each measurement. All the measurements were performed in an air atmosphere at room temperature.

## Simulation of neural networks

The SNNs were composed of 748 input neurons, 200 hidden neurons, and 10 output neurons. Before the training, the MNIST dataset was converted into a time-varying sequence of spikes by transforming the image pixel as a discrete value $X_{ij} \in \{0, 1\}$. In the training, the weighted sum contributed to the neuron membrane potential $U(t)$. When the neuron was sufficiently excited by the weighted sum, the membrane potential reached a threshold $\theta$, and then the neuron emitted a spike to its subsequent connections. The training utilized 60000 training and 10000 test images with a batch size of 200 at each epoch. After 300 epochs, the recognition accuracy was calculated for evaluation.

## DFT calculation

The DFT calculations were performed using the projector-augmented wave method as implemented in the Vienna Ab initio Simulation Package code. The generalized gradient approximation together with the Perdew-Burke-Ernzerhof exchange-correlation function was used. The van der Waals interactions were also included in the calculations using the zero damping DFT-D3 method of Grimme. A uniform grid of $4 \times 4 \times 1$ $k$-mesh in the Brillouin zone was employed to optimize the crystal structures of 1D-S and 1D-R in bulk, $4 \times 4 \times 1$ $k$-mesh for 1D-S and 1D-R slabs, and $1 \times 2 \times 1$ $k$-mesh for SWNTs/1D-S and SWNTs/1D-S interfaces. The interfacial model was built by placing a SWNTs (16,6) on the top of 1D-S or 1D-R (100) surface with a ($5 \times 3$) lateral periodicity. The energy cutoffs of the wavefunctions were set at 450 eV for the bulk and 400 eV for the interfaces. The slab/interface replicas were separated by ~15 Å of vacuum. Each structure was optimized until forces on single atoms were smaller than 0.015 eV/Å.

## Data availability

The experimental data that support the findings of this study are available from the corresponding author upon request.

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

## Acknowledgements

M.Li acknowledges the financial support from the Shenzhen Science, Technology and Innovation Commission (JCYJ20210324131806018) and Research Grant Council of Hong Kong (Project No.PolyU 25301522 and PolyU 15301323), National Natural Science Foundation of China (22373081), and Hong Kong Innovation and Technology Fund (ITS/064/22). J.Y. acknowledges the financial support from Hong Kong Polytechnic University (Grant no. P0042930) and the grant from the Research Grants Council of the Hong Kong Special Administrative Region, China (Project No. PolyU 25300823).

## Author contributions

M.L. and Q.L. conceived the idea. Q.L. prepared samples, devices and performed optical and electrical characterizations. Q.W. and Q.L. performed ultrafast spectroscopy measurements. H.R., L.Z., P.W., and C.W. helped with the sample fabrication and characterizations. J.Y. and Y.Z. performed DFT calculations. All authors agreed with the final version of the manuscript. M.L. led the project.

## Competing interests

The authors declare no competing interests.
