## [Peer Review File · Nature Communications]

REVIEWER COMMENTS

Reviewer #1 (Remarks to the Author):

The paper from Liu et al reports on a novel chiral hybrid-perovskite-chiral SWNT heterostructure. The chiroptical response, excitation dynamics after photoexcitation, synaptic spiking and the application in photonic learning are demonstrated. The paper presents an interesting novel material system, which is carefully and clearly investigated.

I recommend publication after addressing the following suggestions for improvement:

- More information on the rationale for using the 1D hybrid perovskite would be helpful. Could other hybrid perovskites also work for the chiral perovskite-SWNT heterostructures?
- A clearer description of the laser/optical setup that is used for creating the CPL spikes for the PAS operation would be helpful.
- A clearer description of the SNN architecture would be helpful.
- The resolution of the Figures is low and fonts are small, so some labels are hard to read.
- Citations could be added to related recently published work on chiral hybrid perovskites:
[10.1126/sciadv.adh5083](https://doi.org/10.1126/sciadv.adh5083)

Reviewer #2 (Remarks to the Author):

The manuscript by Liu and co-workers describes a heterostructure formed from a chiral halide perovskite material and a carbon nanotube material. This heterostructure is then used to fabricate photonic synapse-like sensor devices with decent sensitivity to circularly-polarized light in the UV region. The authors demonstrate that such sensors can be used for distinguishing and memorizing images with high recognition accuracy in spike neural network simulations.

I find this work a creative approach to leverage the good circular dichroic properties of chiral halide perovskites combined with the comparatively simple device fabrication based on such solution-processable semiconductors (incl. CNTs). While the underlying mechanisms are not surprising or very novel, the successful device demonstration may warrant publication in Nature Communication.

The technical aspects of this work all seem to have been carried out well. I only have a few select comments that should be clarified, namely:

1. The band alignment between halide perovskite and CNT isn't fully clear to me. A hole transfer from the perovskite to the CNT seems to make sense, but what about the alignment of the perovskite conduction band with respect to the CNT: Is electron transfer not also possible?
2. Following from this question, can the authors exclude a FRET-like energy transfer (or perhaps a Dexter-like one even), instead of the proposed hole-only charge transfer?
3. What happens if the authors leave the polarization of their pump beam unchanged at a given handedness but change the probe beam handedness in their TA measurements?
4. In Fig. S5, why is the absorbance, even if showing identical absorbance at the red edge of the spectrum, stronger in the S compared to the R enantiomer in the high-energy region? Given OD is a logarithmic unit, this is really a significant difference and could influence the subsequent data analysis once looking at polarization handedness-dependent characteristics.

- All reviewer comments are displayed in *italics*.
- All our responses are displayed in black.
- Sentences added to the revised manuscript and supplementary information are *blue*.

Response to the reviewers' comments

Reviewer #1

The paper from Liu et al reports on a novel chiral hybrid-perovskite-chiral SWNT heterostructure. The chiroptical response, excitation dynamics after photoexcitation, synaptic spiking and the application in photonic learning are demonstrated. The paper presents an interesting novel material system, which is carefully and clearly investigated.

I recommend publication after addressing the following suggestions for improvement:

Response: We are delighted to hear the reviewer's recognition on the novelty and importance of our research discovery. The reviewer's critical comments are very helpful for us to further improve the manuscript. We have carefully considered the reviewer's comments and made the necessary revision as follows:

1. More information on the rationale for using the 1D hybrid perovskite would be helpful. Could other hybrid perovskites also work for the chiral perovskite-SWNT heterostructures?

Response:

We thank the reviewer for this constructive comment. According to reviewer's suggestion, we have added:

“Here, one-dimensional (1D) chiral halide perovskite is selected due to its high chiroptical performance in the UV region^{34, 35}. SWNT, which has high mobility and is widely used as hole extraction layer in perovskite solar cells^{36, 37}, is chosen as the charge transporting layer in our devices.” on page 6 of revised manuscript with relevant Refs.

Other hybrid perovskites may also work for the chiral perovskite-SWNT heterostructure. To the best of our knowledge, we found a study by Hao *et al.* where they reported a chiral 2D metal halides (R-/S-MBA)₂CuCl₄/SWNT heterostructure for circularly polarized light detection¹. However, this study did not demonstrate or investigate optical synaptic behavior or circularly polarized neuromorphic imaging.

Reference:

1. Hao J, Lu H, Mao L, Chen X, Beard MC, Blackburn JL. Direct detection of circularly polarized light using chiral copper chloride-carbon nanotube heterostructures. *ACS Nano* **15**, 7608-7617 (2021).

2. A clearer description of the laser/optical setup that is used for creating the CPL

spikes for the PAS operation would be helpful.

Response:

We thank the reviewer for raising the concern. According to reviewer’s suggestion, we have revised the sentence as:

“Figure 3a shows the schematic optical setup for circular polarization-dependent synaptic testing (with details of CPL spike generation provided in Supplementary Fig. 25).” on page 11 of revised manuscript.

Moreover, the specification information of utilized polarizers for creating the CPL spikes is added in “Methods” section of the revised manuscript:

“CPL was generated by using a linear polarizer (WP25M-UB), quarter waveplate (AQWP10M-340) and LEDs with different wavelengths purchased from Thorlabs.”

The following **Supplementary Fig. 25** with details is added in the revised Supplementary Information:

Supplementary Fig. 25 | Experimental optical setup for generating the CPL spikes.

An arbitrary function generator, utilized for pulse width modulation, is connected to the LED driver responsible for controlling the LED. When the input voltage reaches the threshold of 5 V, the LED driver switches the output current to the level predetermined by the knob on the front of the unit, indicating that the LED is in the ON state. Conversely, when the input voltage falls below the threshold, the LED remains in the OFF state. After passing through the linear polarizer and quarter waveplate, the light spikes result in the generation of CPL spikes.

3. A clearer description of the SNN architecture would be helpful.

Response:

We thank the reviewer for this constructive comment. According to the reviewer’s suggestion, we added several sentences as shown below for a clearer description of the SNN architecture:

“In the simulation, the input dataset is converted to spikes and transmitted to the first layer, where each neuron corresponds to one pixel of the image dataset. During the feedforward process, the spiking neuron accumulates the weighted sum of input spikes and generates the membrane potential of the neuron. When the input provides sufficient excitation, the membrane potential reaches a threshold (θ), and the neuron releases a spike to its next connections⁴⁶. Additionally, the backpropagation algorithm is employed for weight updates, with the released activation value of the current layer serving as the input for the subsequent layer (see computation process details of the SNN in Supplementary Fig. 29).” on page 16 of the revised manuscript.

The following **Supplementary Fig. 29** with explanations are added in the revised SI:

Supplementary Fig. 29 | Flow diagram of the implementation of the SNN.

Each dataset is encoded as a spike train that conforms to the Poisson equation, and then input to the neurons. The fire rate of neurons relies on the intensity of the corresponding pixel in the dataset image. For an MNIST image, if the spiking probability of a white pixel is 100%, and a black pixel never generates a spike. In the training process, the membrane potential of each output neuron accumulates by the input spikes and weight from other connecting synapses. When the membrane potential value of one output neuron exceeds the threshold, the neuron fires and releases a spike to its next connections, and its membrane potential are reset. The fired neuron will also prevent other neurons from firing by lateral inhibition. Within a period, the neuron keeps refractory state and cannot be fired. Moreover, the corresponding synapses that contributes to the firing result will be strengthened while synapses without contribution for the fire will be weakened. After the training, every category dataset is labeled with fixed threshold and training weights. In the test process, the images are input to the trained network, the output neurons are labeled to the pattern categories in terms of their most firing times to the corresponding input dataset.

4. The resolution of the Figures is low and fonts are small, so some labels are hard to

read.

Response: We sincerely thank the reviewer for this constructive comment. We carefully checked the fonts of all figures emerging in manuscript and supplementary information, and replaced the small font size with an appropriate one. For the low-resolution issue, the resolution of the figures in the manuscript is limited by the submission system, and high-fidelity figures (600 DPI) are uploaded separately, which could be downloaded from the reviewer files in the system.

5. Citations could be added to related recently published work on chiral hybrid perovskites: 10.1126/sciadv.adh5083

Response:

We thank the reviewer for this valuable suggestion. We have added this reference in the revised main text “*These chiral perovskites thus have potential in sensing circularly polarized light (CPL) and differentiating between left-handed CPL (LCP) and right-handed CPL (RCP)*²⁵⁻²⁸.”

Reviewer #2

The manuscript by Liu and co-workers describes a heterostructure formed from a chiral halide perovskite material and a carbon nanotube material. This heterostructure is then used to fabricate photonic synapse-like sensor devices with decent sensitivity to circularly-polarized light in the UV region. The authors demonstrate that such sensors can be used for distinguishing and memorizing images with high recognition accuracy in spike neural network simulations.

I find this work a creative approach to leverage the good circular dichroic properties of chiral halide perovskites combined with the comparatively simple device fabrication based on such solution-processable semiconductors (incl. CNTs). While the underlying mechanisms are not surprising or very novel, the successful device demonstration may warrant publication in Nature Communication.

The technical aspects of this work all seem to have been carried out well. I only have a few select comments that should be clarified, namely:

Response:

We are grateful to hear the reviewer's recognition on the novelty and importance of our research discovery. The reviewer's critical comments are very helpful for us to further improve the manuscript. We have carefully considered the reviewer's comments and made the necessary revision with replies as follows:

1. The band alignment between halide perovskite and CNT isn't fully clear to me. A hole transfer from the perovskite to the CNT seems to make sense, but what about the alignment of the perovskite conduction band with respect to the CNT: Is electron transfer not also possible?

2. Following from this question, can the authors exclude a FRET-like energy transfer (or perhaps a Dexter-like one even), instead of the proposed hole-only charge transfer?

Response: We sincerely thank the reviewer for the valuable comment. Considering the fact that comment 1&2 are regarding the same concern of reviewer, we thus answered as shown below.

Previous literature, our UPS and newly added TA results all well support the hole transfer from H-PVK to SWNTs rather than energy transfer, and the long-lived holes are existing in the SWNTs of our heterostructures.

(i) Despite the fact that perovskite bandgap energy (including our chiral perovskites) is typically larger than that of single walled carbon nanotubes (SWNTs), no FRET energy transfer (requiring dipole-dipole interaction) or Dexter-like energy transfer (requiring wavefunction overlap) from perovskite film to SWNTs was reported. Since FRET or Dexter-like energy transfer requires the donor and acceptor very close $\sim 1-2$ nm, which is usually occurs between quantum dot/SWNTs². In our perovskite/SWNTs heterostructure, the perovskite film is ~ 70 nm (Fig. S3), therefore, FRET or Dexter-like energy transfer from perovskite film to SWNTs should not be efficient.

(ii) SWNTs has been widely used in hole transport layer of perovskite solar cells to improve the device performance³⁻⁷. SWNTs has a high hole mobility of tens of $\text{cm}^2 \text{V}^{-1}$

1 s^{-1} . Previous report has demonstrated the ground-state electron transfer from perovskite to SWNTs, which induces the energy band bending for the photoexcited holes transfer to SWNTs⁸. Our UPS results also support the hole transfer rather than electron transfer. To make it clearer to reader, we thus added the following figure as Fig. S13 with relevant explanation below the figure in the revised Supplementary Information, and replaced the Fig. 1c with the right figure of Fig. S13.

Supplementary Fig. 13 | Band alignments of H-PVK and SWNTs before and after connection. (E_{vac} is the vacuum energy level, E_F is the Fermi energy level).

Once H-PVK and SWNTs are connected, ground-state electron transfer occurs from PVK (with a low work function, higher E_F , from UPS) to SWNTs (with a high work function, lower E_F). This transfer continues until the Fermi levels align across the interface under equilibrium conditions, and results in interfacial energy band bending. Consequently, an electron depletion region can form at the H-PVK side, creating a potential barrier of 0.7 eV. This potential barrier is larger than the conduction band offset of 0.4 eV, making it unfavorable for photoexcited electron transfer from H-PVK to SWNTs.

And revised the following sentence on page 7 of the revised manuscript:

“Upon contact between H-PVK and SWNTs, an interfacial band bending with a potential barrier of 0.7 eV is formed (Fig. 1c), which can repel the photoexcited electrons and facilitate the transfer of photoexcited holes from H-PVK to SWNTs (see details in Supplementary Fig. 13). More evidence of photoexcited charge separation are provided in the following section.”

(iii) To further verify the hole transfer to SWNTs, we performed near-IR TAS measurements (Figure S22). A photobleaching (PB) band with peak at $\sim 1050 \text{ nm}$ arising from the stating filling of S_{11} exciton transition (Fig. S22a) along with a fast decay lifetime ($< 1 \text{ ps}$) is observed in our SWNTs sample, which is consistent with previous results⁹⁻¹¹. After addition of our chiral perovskites, an additional long-lived decay component probed at 1050 nm is observed in heterostructures, which could arise from the stating filling of the separated holes. These observations are consistent with

previous reports of separated long-lived holes in SWNTs^{10, 12}.

Supplementary Fig. 22 | The near-IR transient absorption measurement. (a, c, e) TA spectra at various delay time after subtraction of the background signal and (b, d, f) normalized photobleaching dynamics probed at ~ 1050 nm for SWNTs, 1D-R/SWNT, and 1D-R/SWNTs, respectively. The excitation wavelength is 340 nm.

The photobleaching (PB) band with peak at ~ 1050 nm arising from the stating filling of S_{11} exciton transition along with a fast decay lifetime (< 1 ps) is observed in our SWNTs sample (Supplementary Figs. 22a-b), which is consistent with previous results¹⁻³. After addition of our chiral perovskites, an additional long-lived decay component probed at 1050 nm is observed in heterostructures (Supplementary Figs. 22c-f), which arise from the stating filling of the separated holes. These observations are consistent with previous reports of separated long-lived holes in SWNTs^{2, 4}.

Note that the fast decay from S_{11} exciton is still existing in TA dynamics of heterostructures, which could be due to the energy transfer and/or the direct excitation of SWNTs by the pump laser. However, even if there are some short-lived excitons in SWNTs, they cannot contribute to the photocurrent due to the fast recombination, it thus could only affect the quantum efficiency of device, and won't affect the fundamental

concept and function of our LCP/RCP light resolvable artificial synapse devices.

The above **Supplementary Fig. 22** and related explanation are added in the revised supplementary information.

Furthermore, the following sentence:

“The near-infrared (IR) TA is used to probe the carrier population dynamics in SWNTs. A long lived PB band at ~ 1050 nm is observed in H-PVK/SWNTs, which is attributed to the photoexcited hole-transfer from H-PVK to SWNTs (see details in Supplementary Fig. 22).” has been added on page 10 of the revised manuscript.

(iv) Importantly, we also performed the change of TA signal ($\Delta\Delta A$) probed at the SWNTs PB band as function of pump laser spike numbers (similar to the Fig. 2g in the previous version of manuscript, which is Supplementary Fig. 23 in the revised SI). As shown below in Fig. 2g in the revised manuscript, there is no change of $\Delta\Delta A$ due to the fast recombination of electron and hole occurs inside the SWNTs alone. In contrast, in the H-PVK/SWNTs, $\Delta\Delta A$ of 1D-S/SWNTs shows an increasing trend with the increasing pump spike numbers, which thus indicate the photoexcited long-lived holes are existed in the SWNTs of the heterostructures and contribute to the constructing artificial synapses. The following Fig. 2g has been added with relevant explanations on page 10 in the revised manuscript.

Fig. 2g Laser spike-numbers dependent change of TA amplitudes probed at ~ 1050 nm and 1ps of SWNTs (upper panel) and 1D-S/SWNTs (lower panel) under 340 nm LCP/RCP excitation.

Lastly, note that the fast decay from S_{11} exciton is still exist in TA dynamics of heterostructures, which could be due to the energy transfer and/or the direct excitation of SWNTs by the pump laser. Therefore, we cannot say there is only the hole-transfer (which is also not mentioned in the manuscript). However, even if there are some energy transfer from perovskite to SWNT, which can result in the short-lived excitons in SWNTs but without contribution on photocurrent, it thus could only affect the quantum efficiency of device, and won't affect the fundamental concept and function of our LCP/RCP resolvable artificial synapse devices.

Reference:

2. Shafran E, Mangum BD, Gerton JM. Energy Transfer from an individual quantum dot to a carbon nanotube. *Nano Lett.* **10**, 4049-4054 (2010).
3. Aitola K, et al. Carbon nanotube-based hybrid hole-transporting material and selective contact for high efficiency perovskite solar cells. *Energy Environ. Sci.* **9**, 461-466 (2016).
4. Habisreutinger SN, Noel NK, Larson BW, Reid OG, Blackburn JL. Rapid charge-transfer cascade through SWCNT composites enabling low-voltage losses for perovskite solar cells. *ACS Energy Lett.* **4**, 1872-1879 (2019).
5. Habisreutinger SN, Leijtens T, Eperon GE, Stranks SD, Nicholas RJ, Snaith HJ. Enhanced hole extraction in perovskite solar cells through carbon nanotubes. *J. Phys. Chem. Lett.* **5**, 4207-4212 (2014).
6. Liu C-K, et al. High-performance quasi-2D perovskite/single-walled carbon nanotube phototransistors for low-cost and sensitive broadband photodetection. *Small Struct.* **2**, 2000084 (2021).
7. Blackburn JL. Semiconducting single-walled carbon nanotubes in solar energy harvesting. *ACS Energy Lett.* **2**, 1598-1613 (2017).
8. Schulz P, Dowgiallo A-M, Yang M, Zhu K, Blackburn JL, Berry JJ. Charge transfer dynamics between carbon nanotubes and hybrid organic metal halide perovskite films. *J. Phys. Chem. Lett.* **7**, 418-425 (2016).
9. Huang L, Pedrosa HN, Krauss TD. Ultrafast ground-state recovery of single-walled carbon nanotubes. *Phys. Rev. Lett.* **93**, 017403 (2004).
10. Kang HS, et al. Long-Lived Charge Separation at heterojunctions between semiconducting single-walled carbon nanotubes and perylene diimide electron acceptors. *J. Phys. Chem. C* **122**, 14150-14161 (2018).
11. Graham MW, et al. Exciton Dynamics in Semiconducting Carbon Nanotubes. *The J. Phys. Chem. B* **115**, 5201-5211 (2011).
12. Ellingson RJ, et al. Ultrafast photoresponse of metallic and semiconducting single-wall carbon nanotubes. *Phys. Rev. B* **71**, 115444 (2005).

3. What happens if the authors leave the polarization of their pump beam unchanged at a given handedness but change the probe beam handedness in their TA measurements?

Response:

We thank the reviewer for this constructive comment.

According to the reviewer's suggestions, we performed TA measurements with the polarization of the pump beam unchanged at a given handedness but change the probe beam handedness (Fig. R1 & R2 for 1D-S and 1D-R, respectively).

Note that chirality may result in spin splitting of electronic bands in chiral perovskites (see Fig. R1 inset). Therefore, by using LCP/RCP probe light and following the optical selection rules, we can monitor the carrier dynamics with different spin states. Furthermore, the polarization degree of TA signal related to spin-splitting can be quantified by the following equation

$$P(t) = \frac{\Delta A_{\text{counter}} - \Delta A_{\text{co}}}{\Delta A_{\text{count}} + \Delta A_{\text{co}}}$$

where $\Delta A_{\text{counter}}$ and ΔA_{co} are the TA signals pumped and probed with counter-polarized and co-polarized configurations.

As shown in Fig. R1, under 350nm LCP excitation for 1D-S sample (as it has a higher absorption for LCP light, so here LCP is used), there are not obvious difference for LCP and RCP probed TA spectra (Fig. R1a). And also, small P of $\sim 3\%$ is observed, which indicate that chirality induced spin-splitting is weak in our 1D-S, may be due to the strong electron-photon interaction. There is also no obvious difference in TA spectra under LCP or RCP probe, which indicate that the spin-polarized charge transfer to SWNT is nearly same. For 1D-R sample, a higher polarization of $\sim 10\%$ is observed, which is similar to the previous observation of TA signal polarization of chiral perovskites^{13, 14}.

It should be noted that, in our TA dynamics in the manuscript, LCP light is used to pump sample as 1D-S (1D-R) samples has a stronger absorption in LCP (RCP) light. And linearly polarized (LP) probe light is used, as linearly polarized light is superposition of LCP and RCP light, thus LP light can favorably eliminate the impact of potential chirality induced spin polarization, and probe all transfer holes to SWNTs.

As the difference of spin polarization between 1D-S and 1D-R is not the scope of this work, these data of Fig. R1 & R2 are not relevant to this work, so they are not being provided in the Supplementary Information. However, readers still can see these data as peer reviewed file will be uploaded after manuscript publication.

Figure R1. (a) Circularly polarized TA spectra of 1D-S under LCP light (σ^-) pump, σ^- probe (black) or RCP light (σ^+) probe (red) of 1D-S. (b) Polarization as a function of delay time of bandage PB signal in 1D-S. Inset: schematic diagram of the continuum bands with spin splitting in chiral perovskites, and related optical transitions with circularly polarized light. (c) Circularly polarized TA spectra of 1D-S/SWNTs.

Figure R2. (a) Circularly polarized TA spectra of 1D-R under RCP light (σ^+) pump, σ^+ probe (black) or LCP light (σ^-) probe (red) of 1D-R. (b) Polarization as a function of delay time of bandage PB signal in 1D-R. Inset: schematic diagram of the continuum bands with spin splitting in chiral perovskites, and related optical transitions with circular polarized light. (d) Circularly polarized TA spectra of 1D-R/SWNTs.

Reference:

13. Liu S, et al. Optically induced long-lived chirality memory in the color-tunable chiral lead-free semiconductor (R)/(S)- $\text{CHEA}_4\text{Bi}_2\text{Br}_x\text{I}_{10-x}$ ($x = 0-10$). *J. Am. Chem. Soc.* **144**, 14079-14089 (2022).
14. Liu S, et al. Bright circularly polarized photoluminescence in chiral layered hybrid lead-halide perovskites. *Sci. Adv.* **9**, eadh5083 (2023).

4. In Fig. S5, why is the absorbance, even if showing identical absorbance at the red edge of the spectrum, stronger in the S compared to the R enantiomer in the high-energy region? Given OD is a logarithmic unit, this is really a significant difference and could influence the subsequent data analysis once looking at polarization handedness-dependent characteristics.

Response:

We appreciate the reviewer's comment, and we apologize for the oversight of not properly labeling the absorbance measurement condition in Figure S5, which led to the misunderstanding. In Figure S5, the observed difference in absorbance is due to the utilization of left-handed circularly polarized light (LCP) by the CD spectrometer during the measurement. As a result, when LCP is used for absorbance measurement, it leads to identical absorbance values at the red edge of the spectrum below bandgap, but a stronger absorbance is observed for the S enantiomer compared to the R enantiomer in the high-energy region above bandgap.

In addition, we measured the linear optical absorbance of S and R enantiomer-based samples with UV-Vis spectrometer. As shown in Figure R3, the S-type and R-type heterostructures show no obvious difference in linear absorption spectra, and this result also demonstrates that the absorbance difference in Figure S5 is caused by the measurement light source. The related measurement description has been added on page 6 of the revised supplementary information.

Figure R3. Linear absorbance spectra of H-PVK/SWNTs measured by UV-Vis spectroscopy with unpolarized light source.

The new caption is updated for **Figure S5** in supplementary information:

“**Supplementary Fig. 5.** Absorbance spectra of H-PVK/SWNTs measured under LCP condition.”

Furthermore, it would to be noted that “*polarization handedness-dependent characteristics*” (e.g. LCP/RCP dependent photocurrent, responsivity, and data in Fig. 3-5 from sample 1D-S) depend on the absorption difference of LCP and RCP light (i.e., the CD, determined as $CD = 3298.2(\epsilon_L - \epsilon_R)$ and reported by degrees in Fig. 1d, where ϵ_L and ϵ_R are the molar extinction coefficient difference for LCP and RCP light) of one type sample (either 1D-S or 1D-R).

Therefore, even if there may have some variation in absorbance between the 1D-S and 1D-R samples, which could arise from differences in film thickness or concentration, it would not impact the CD value or the polarization handedness-dependent characteristics and analysis for either the 1D-S or 1D-R sample.

REVIEWERS' COMMENTS

Reviewer #1 (Remarks to the Author):

The authors have addressed and answered my comments.

Reviewer #2 (Remarks to the Author):

I am very pleased with the revised version of this manuscript and appreciate the added work put in by the authors to improve accessibility to the reader. I now recommend publication.

- All reviewer comments are displayed in *italics*.
- All our responses are displayed in black.

Response to the reviewers' comments

Reviewer #1

The authors have addressed and answered my comments.

Response: We sincerely appreciate the reviewer's time and effort in reviewing our manuscript. We are grateful for their recommendation to publish our research.

Reviewer #2

I am very pleased with the revised version of this manuscript and appreciate the added work put in by the authors to improve accessibility to the reader. I now recommend publication.

Response: We sincerely appreciate the reviewer's time and effort in reviewing our manuscript. We are grateful for their recommendation to publish our research.